# Speeding up Policy Optimization with Vanishing Hypothesis and Variable Mini-Batch Size

## Abstract

Reinforcement learning-based algorithms have been used extensively in recent years due to their flexible nature, good performance, and the increasing number of said algorithms. However, the largest drawback of these techniques remains unsolved, that is, it usually takes a long time for the agents to learn how to solve a given problem. In this work, we outline a novel method that can be used to drastically reduce the training time of current state-of-the-art algorithms like Proximal Policy Optimization (PPO). We evaluate the performance of this approach in a unique environment where we use reinforcement learning to help with a practical astronomical problem: where to place a fixed number of observatory stations in the Solar System to observe space objects (e.g. asteroids) as permanently as possible. That is, the reward in this scenario corresponds to the total coverage of the trajectories of these objects. We apply noisy evaluation for calculating the reward to speed up the training, which technique has already been efficiently applied in stochastic optimization. Namely, we allow the incorporation of some additional noise in the reward function in the form of a hypothesis term and a varying mini-batch size. However, in order to follow the theoretical guidelines, both of them are forced to vanish during training to let the noise converge to zero. Our experimental results show that using this approach we can reduce the training time remarkably, even by 75%.

## 1 Introduction

Reinforcement learning (RL) was in the focus of numerous studies in the last few years. As such, plenty of new algorithms have emerged and are continuously surfacing that improve the already existing RL algorithms or combine the traditional methods with modern deep learning techniques, like deep Q-learning (Mnih et al., 2013), Proximal Policy Optimization (PPO) (Schulman et al., 2017), Asynchronous Advantage Actor-critic (Mnih et al., 2016b) or Soft Actor-Critic (Haarnoja et al., 2018). Researchers have used these algorithms with great success, solving various challenging problems, ranging from using RL to play table games like Go (Silver et al., 2016; 2017), playing Massively Multiplayer Online (MMO) games (Suarez et al., 2019), to solving complex problems in robotics (Plappert et al., 2018). Recent research (Reed et al., 2022) has also shown that we can train a single RL agent in a way that it can be used to solve different tasks, indicating the high generalization capabilities of these types of algorithms. However, one of the biggest problems of these algorithms – especially on-policy methods – still remains: the optimization process, and hence the training of the agent, takes a really long time (Yarats et al., 2021; Yu, 2018). This is especially true in the case of policy-based algorithms which are notorious for being sample inefficient (Bastani, 2020). Although PPO, which is also a policy-based method, has better sample complexity (Schulman et al., 2017) than the original policy gradient algorithm (Mnih et al., 2016a), it still suffers from this phenomenon. Despite this drawback, PPO is still widely used due to its convergence to at least a local optimum, in contrast to value-based methods, where convergence is not necessarily guaranteed, which phenomenon is referred to as the "deadly triad" by Sutton & Barto (2018). In this work, we show that we can remarkably reduce the training time of a PPO agent while still preserving its convergence and achieving the same cumulative rewards by treating the loss as an energy function and incorporating some noises in the optimization process.

In real-world applications it is quite common that the energy function cannot be evaluated precisely because of certain noise corruptions. If the noise is too large then a meaningful optimization cannot

take place. On the other hand, we can also inject some corruption deliberately to speed up the optimization process via applying noisy evaluation (Gelfand & Mitter, 1989). The original idea was to perform only a rough evaluation which approach can save a lot of computational time if the energy function is complex, and its proper evaluation is laborious. It has already been shown that this approach is valid only in the case when the corruption vanishes as the search progresses to be able to keep up the convergence behavior of the original (uncorrupted) model. Specifically, for simulated annealing (SA) Gelfand & Mitter (1989) have shown in what extent the noise should converge to 0 to preserve the search efficiency of the uncorrupted case.

In this work, we focus on policy-based approaches and further refine the noisy evaluation idea to integrate it in the RL framework in different ways. The first integration is possible only in such applications – including ours –, where a batch size is used during the gradient descent or ascent update. This is true for policy gradient algorithms which measure the goodness of a policy $\pi$ as the cumulative reward achieved in a set time frame via $r(\pi) = \sum_t [r(s, a, s')|s = s_t, a = a_t, s' = s_{t+1}]$

for each timestep $t \in \mathbb{N}$ in the given time frame and use a gradient estimate to update $\pi$. In such cases we can apply the well-known mini-batch approach with including only a part of these timesteps in one update cycle. However, opposite to the traditional mini-batch approach that keeps the batch size constant, we also consider it as a noise w.r.t. a full-batch validation. Consequently, we also follow the above requirement to vanish the noise, which can be achieved here with continuously increasing the batch size during training till reaching the total number of observations in the episode.

Beyond considering variable mini-batch size, we also include a hypothesis term $h(s')$ for every subsequent state $s'$ in the reward function $r(s, a, s')$ as a direct feedback for our agent. This procedure can also be interpreted as adding noise in the light of the above summary. Our intention with this addition is to speed up the training process and improve the sample efficiency, as we believe that an appropriately formulated hypothesis can direct the search to the optimal location and hence make the convergence faster. However, we can never be sure that our hypothesis indeed has this behaviour, since we can be mistaken with it. Moreover, to take the above considerations also into account, the noise term should tend to 0 to keep up the convergence behavior of the original RL model. Thus, the hypothesis term will be added in such a way that it will vanish as the search progresses. As we will see, both the variable mini-batch and the vanishing hypothesis term are able to speed up the training process, however their simultaneous integration will improve it further.

The rest of the paper is organized as follows. In section 2 we properly exhibit how the hypothesis term and the variable mini-batch can be integrated in a reward function considered in policy optimization. Our special application domain that originally motivated the research is introduced in section 3. Then, section 4 presents the details of the proper implementation. In section 5 we show our experimental results suggesting a meaningful speed up in finding the solution for our problem. Finally, some conclusions are drawn in section 6.

## 2 METHODOLOGY

Policy gradient methods like PPO use an estimation of the gradient to update their policy $\pi$ parameterized by a set of weights $\theta$ ($\pi_\theta$ for short) so that the expected cumulative reward

$$\mathbb{E}_{\pi_\theta}[\sum_t r_t | r_t = r(s, a, s'), s = s_t, a = a_t, s' = s_{t+1}] \tag{1}$$

increases. To this end, they use an estimation similar to the following expression (Schulman et al., 2017) for each timestep $t$ and perform gradient ascent:

$$\widehat{g} = \widehat{\mathbb{E}}_t[\nabla_\theta \log \pi_\theta(a_t|s_t)\widehat{A}_t], \tag{2}$$

where $\widehat{A}_t$ is the estimation of the advantage. PPO also uses a clipped surrogate objective to prevent large updates during the optimization process. Although this facilitates training to a certain degree, making PPO converge faster than the vanilla policy gradient method (Mnih et al., 2016a) and achieve higher scores as shown in Schulman et al. (2017), the training still takes a long time. What is even more important is the fact that in the PPO paper it can also be observed that in several environments the agents achieved really low scores in a significant chunk of the total training time before finally

converging to an optimum. This raises the question whether this part of the training could be further improved to reduce the total time required for training the agent.

In our work, we focus on reducing this time by incorporating some additional noise and performing noisy evaluation during the training process, which technique has been shown to work well with regular stochastic optimization problems. Namely, for SA it has been shown in Gelfand & Mitter (1989) that a normally distributed noise with mean 0 and variance $\left(\sigma^{(k)}\right)^2 > 0$ in the $k$-th iteration still let the search converge to the globally optimal solution with probability 1 if

$$\sigma^{(k)} = o\left(T^{(k)}\right). \tag{3}$$

Besides the temperature, an alternative constraint has also been formulated in Gutjahr & Pflug (1996) for the same purpose for the number of iterations as

$$\sigma^{(k)} = O\left(k^{-\beta}\right) \text{ with some } \beta > 1. \tag{4}$$

Though the conditions (3), and (4) require the variance of the noise to decrease along with the progress of the exploration, it is obvious that the noise should also vanish to preserve convergence.

A very attractive expectation towards any optimization methods is that the applied heuristics that is used to speed up the search should guarantee the original convergence properties; esp. to preserve convergence to the global optimum. Though numerous very efficient strategies have been proposed to improve exploration, unfortunately, the theoretical preservation of the convergence has been rarely proved and even in such cases strict limitations had to be applied. For example, even for the original SA algorithm it has been shown (Geman & Geman, 1984) that only the logarithmic cooling schedule can preserve global convergence, while in practice the linear, and the exponential schemes became very popular. Regarding our current study, we can have very similar claims for the PPO algorithm. As a relatively new approach, in spite of its very popular and well-applicable nature, the theoretical foundations have already just begun recently (Bhandari & Russo, 2019; Agarwal et al., 2021). Thus, regarding the above discussion, we will follow the theoretical suggestions for noisy evaluation to incorporate our speed-up considerations as noises in the PPO algorithm. Accordingly, they should vanish by the end of the training to fulfill (3), and (4).

## 2.1 VARIABLE MINI-BATCH SIZE

The first approach that we have used to incorporate noise in the training process relies on the number of samples used in the rollout during the gradient update. For this to work, the most important requirement is to have a batch of samples to work with. Therefore, we will use the rollout itself and $\widehat{\mathbb{E}}_t$ from (2) which is the average over a batch of samples, but instead of defining a static, fixed batch size $K$, we will start from a small $K$ and gradually increase it up until the total number of observations $N$.

The batch size $K$ can be changed with different dynamics for the $k$-th iteration step. In Tóth et al. (2020) it has been proved that the variance of the noise coming from this sampling should fulfill

$$\sigma_K^{(k)} \gtrsim T^{(k)}(1 - \epsilon)^k, \quad 0 < \epsilon \ll 1 \tag{5}$$

using SA to preserve convergence. Accordingly, it was also derived that

$$K \approx \frac{N\sigma_{max}^2}{(N - 1)\sigma_K^{(k)2} + \sigma_{max}^2} \tag{6}$$

should hold for the minimal batch size at the $k$-th iteration step in SA, where $\sigma_{max}$ is the worst-case maximum value of the whole population standard deviation.

The trend to increase $K$ depends on the cooling profile selected in SA to calculate the temperature $T^{(k)}$ for the $k$-th iteration step in (5). The linear cooling schedule

$$T^{(k)} = T^{(0)} - \alpha k, \quad 0 < \alpha \tag{7}$$

leads to the requirement

$$\sigma_K^{(k)} \approx \left(T^{(0)} - \alpha k\right)(1 - \epsilon)^k, \quad 0 \leq \alpha \leq 1, \, 0 < \epsilon \ll 1 \tag{8}$$

by applying (5) and also to

$$K = \frac{N\sigma_{max}^2}{(N-1)(T^{(0)} - \alpha k)^2(1-\epsilon)^{2k} + \sigma_{max}^2} \tag{9}$$

by (6). Similarly, for the exponential schedule

$$T^{(k)} = T^{(0)}\,\alpha^k \text{ with } 0 \le \alpha \le 1, \tag{10}$$

we have

$$\sigma_K^{(k)} \approx T^{(0)}\,\alpha^k(1-\epsilon)^k \text{ with } 0 \le \alpha \le 1, \text{ and } 0 < \epsilon < 1, \tag{11}$$

and

$$K = \frac{N\sigma_{max}^2}{(N-1)\left(T^{(0)}\,\alpha^k(1-\epsilon)^k\right)^2 + \sigma_{max}^2} \tag{12}$$

to meet the requirements.

Similarly to various SA applications, we have found the trend (12) corresponding to the exponential cooling schedule the most efficient in our experiments to adjust $K$. To incorporate the variable mini-batch size, we slightly modify the loss function described in Schulman et al. (2017) and arrive at the following formula for the loss function $L(\theta)$:

$$L(\theta) = \frac{1}{K}\sum_{t=0}^{K} \left[ \min(R_t(\theta)\widehat{A}_t, clip(R_t(\theta), 1-\varepsilon, 1+\varepsilon)\widehat{A}_t) \right], \tag{13}$$

where $R_t(\theta)$ is the probability ratio calculated between the current ($\pi_\theta$) and the old ($\pi_{\theta_{old}}$) policies, and is defined as in the original paper as:

$$R_t(\theta) = \frac{\pi_\theta(a_t|s_t)}{\pi_{\theta_{old}}(a_t|s_t)}. \tag{14}$$

## 2.2 HYPOTHESIS AS NOISE

The second approach that we have realized is using a hypothesis as noise. To this end, we have decided on a fixed hypothesis $h : \mathcal{S} \to \mathbb{R}$, where $\mathcal{S}$ is the state space that we have incorporated in the reward function $r(s, a, s')$ in the following way. For each timestep $t$ within the $k$-th iteration, when the agent used an action $a_t$ in a state $s_t$ and arrived at the next state $s_{t+1}$, it received the reward $r_t = r(s_t, a_t, s_{t+1})$ through the formula

$$r(s_t, a_t, s_{t+1}) = \left(1 - \frac{\sigma_K^{(k)}}{\max_{s\in\mathcal{S}}(h(s))}\right)\widehat{r}_t + \frac{\sigma_K^{(k)}}{\max_{s\in\mathcal{S}}(h(s))}h(s_{t+1}), \tag{15}$$

where $\widehat{r}_t$ denotes the traditional reinforcement learning reward that the agent would receive without taking any hypothesis into account, $h(s')$ is the reward for arriving in the next state $s' = s_{t+1}$ according to our hypothesis $h$, and $\max_{s\in\mathcal{S}}(h(s))$ is the theoretically obtainable maximal reward at a timestep. The role of the trade-off parameter $\sigma_K^{(k)}/\max_{s\in\mathcal{S}}(h(s))$ in (15) is to balance the final reward $r(s, a, s')$ between the traditional and hypothesis-based rewards. Moreover, with this selection of the trade-off we incorporate the hypothesis as a vanishing noise which follows the same trend as the variable mini-batch size. In this way we preserve the original convergence behavior of PPO and by using the trade-off parameter to weight both types of rewards we also make sure that the final reward $r(s, a, s')$ is on the same scale as the traditional reward $\hat{r}_t$ to make them comparable.

The adjustments of the parameters discussed throughout this section are task-dependent and will be given properly in section 4.

## 3 APPLICATION DOMAIN

Our practical task motivating the research focuses on optimal sensor placement. The goal is to place sensors in a spatial domain in such a way that we achieve the maximum result according to a particular reward function. This task has received a great interest in recent years as a RL problem.

Research has been carried out on a wide variety of topics, such as placing sensors in greenhouses to measure humidity and temperature (Uyeh et al., 2021), but there have also been publications generalizing the idea (Wang et al., 2019). Perhaps the greatest scientific acclaim of all, however, came from the Google Brain team, who in Mirhoseini et al. (2017); Goldie & Mirhoseini (2020) presented a RL algorithm that solved the chip placement dilemma significantly faster and better than human experts.

By studying the approaches that set out to solve this type of task, we can observe a broadly consistent methodology for dealing effectively with the optimal placement problem. The solution starts with the transformation of the problem into a Markov decision process. Typically, the environment is thought of as a discrete mapping of the domain, so the problem can be formulated in general terms as follows: the goal is to place $M$ sensors in a finite number of possible locations to provide optimal coverage. In these problems, the agent is the entity that chooses the actions that usually place or move a sensor. The reward function varies from problem to problem.

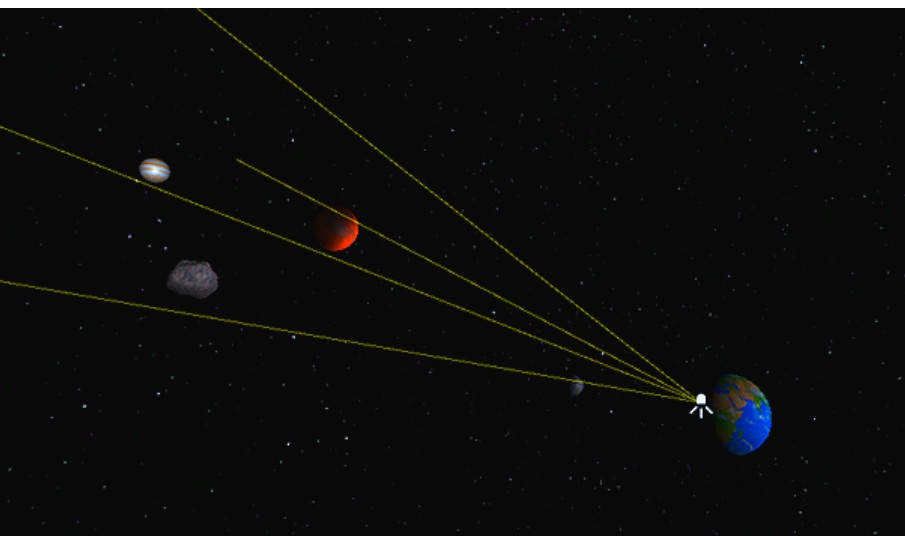

Figure 1: Observing space objects in the Solar System from the Earth.

We investigate the applicability of RL in a time-varying continuous model with geometric coverings. Specifically, it strives to find out how to optimally position observatories on the surface of the Earth, so they provide the maximum possible coverage for monitoring the celestial bodies of the Solar System for a chosen time interval (see Figure 1). Specific subtasks may relate to the fast detection and observation of the trajectories of space objects e.g. asteroids or unidentified ones. Generally speaking, the problem at hand is so complex that with theoretical models it is virtually impossible to handle. However, the presented RL approach is able to address it and can be extended to even more complex problems. One simple extension can be e.g. to let the observatories to be installed also on other planets/moons of the Solar System.

To achieve optimal placement, the software uses the methodology described above. In our case, this means that at each step a location is chosen for an observatory. It is achieved by generating two values from $[-1, 1]$, where the first represents latitude mapped to $[-90, 90]$, and the second means longitude mapped to $[-180, 180]$. Although giving partial rewards after each step seemed reasonable, it did not help the learning process. When all the observatories are placed, rewards are calculated based on coverage earned, and the episode ends.

The solution was implemented in Unity (Haas, 2014), using the open-source Pytorch-based Unity ML-Agents Toolkit (Juliani et al., 2018). The main advantage of this framework is that it can deal exceptionally well with our geometry-based problem, while also having ease of implementation through the ML-Agents toolkit. On top of that, it provides a well-optimized pre-implemented version of PPO, the algorithm most often used to solve positioning problems recently.

## 4 IMPLEMENTATION

### 4.1 DATA ACQUISITION

To find the optimal sensor placement, it is essential to have the location of the celestial objects of the Solar System at different time instants to handle occlusions. For this task, the DE 421 planetary and lunar ephemeris (Folkner et al., 2009) was used, which contains estimates for the orbits of all objects from 1900 to 2050; our examples will fall in this time interval.

In ephemerides, the positions of the celestial bodies are typically described by three values, Alpha, Delta (angles), and distance from a given center (origin). In astronomy, different reference frames are used for different problems. The frame we have chosen for this task is the International Terrestrial Reference System (ITRS) (Petit & Luzum, 2010). ITRS is a coordinate system whose origin is the Earth's center of mass, and as the Earth rotates, so does the coordinate system itself. It also identifies the location of points with 3D coordinates ($X, Y, Z$ triplets), with the $XY$ plane going through the Earth's equator and the $Z$-axis pointing towards the North Pole, which makes it ideal for working with Unity.

The coordinates of the ephemerides represent the actual positions of the celestial bodies in space. However, the task requires information about where a given celestial body appears in the sky when observed from Earth. This conversion was done with the Skyfield Python package Rhodes (2019), designed to easily obtain the positions of different planets and celestial bodies. The procedure to convert the values obtained from DE421 to ITRS is as follows:

- Selecting the Earth's center of mass as the observation point.
- Selecting a target to observe.
- Generating the astrometric position, which considers the effect of light travel time.
- Generating the apparent position, which applies additional effects that might affect where celestial objects appear in the sky.
- Converting the position to ITRS.

### 4.2 LEARNING PROCESS

For the implementation to address multiple problems, a flexible framework has been developed that can be parameterized with different data. These parameters include information about the observatories and celestial bodies, which were collected into a CSV file with a structure of:

- 1st row: Angle of View (AOV) of the observatories to be placed. For example, '30,20,10' means that three observatories are placed, with AOVs of $30°$, $20°$, and $10°$, respectively.
- 2nd row: Diameter of the bodies to be observed. As with the AOV, the number of values displayed is the same as the number of celestial bodies being monitored. Celestial bodies are represented by a regular sphere.
- 3rd row: The importance value of the objects to be observed (used for reward calculation). For example, if we rewarded closer objects to the Earth more, we could have an importance value based on distance.
- 4th and the following rows: Observations in ITRS reference frame in '$X$ / $Y$ / $Z$' format. A row represents an observation at a time instant, with a column describing the position of the celestial body. For example, on January 1, 2000, the position of the Sun after scaling is: -135.32690109260457 / -57.646292187056936 / -1.7959018771790942

At the beginning, the only entity in the environment is the Earth, since it is the only object, whose position will not change during the learning process (no need to move or rotate it thanks to the reference system). Solving the task starts with processing the parameters. We read the problem to be solved and save the data related to the observatories and celestial bodies. Then, we create the corresponding spheres representing these objects in the environment at the appropriate size. The learning process is as described above, with a new observatory being placed at each step. After all the observatories are placed, the reward $\hat{r}_t$ in (15) is calculated, and the episode ends. The whole process can be described more formally by Algorithm 1.

---

**Algorithm 1** Optimal placement of observatories

---

 1: **Input**
 2:     genPos        generated positions
 3:     obs           observatories to be placed
 4:     planets       planets to be observed
 5:     maxPoints     maximum achievable points based on importance values

---

 6: **Output**
 7:     reward        reward earned

---

 8: $reward \leftarrow 0$
 9: **for each** $gp \in genPos$ **do**
10:     $distinctPlanetsSeen \leftarrow \emptyset$
11:     position $planets$ according to $gp$
12:     **for each** $o \in obs$ **do**
13:         $planetsInCone \leftarrow$ all planets that are inside an $o.angle$ cone
14:         **for each** $p \in planetsInCone$ **do**
15:             send ray from $o$ to $p$
16:             **if** ray collides first with $p$ and $distinctPlanetsSeen \cap p = \emptyset$ **then**
17:                 $distinctPlanetsSeen = distinctPlanetsSeen \cup p$
18:             **end if**
19:         **end for**
20:         $reward + = getPointsBasedOnPlanets(distinctPlanetsSeen)$
21:     **end for**
22: **end for**
23: **return** $reward/(len(genPos) * maxPoints)$

---

The way it is done is by going through all the generated positions from the CSV file and checking the number of distinct objects seen by that placement. A celestial body is considered seen, if it is within the observatory's AOV, and is not obscured by another object. The reward earned in a given generated position is the sum of the importance values assigned to the objects seen. The sum of the rewards earned by the placement is divided by the number of generated positions and the maximum achievable points, so it is always less than 1.0 (Unity ML-Agents best practice).

Our main example considers placing $M = 3$ observatories, each with an AOV of $30°$, trying to find the optimal placement for the time interval from 01.01.2022 to 12.12.2032 with four daily samples, at 0 am, 6 am, 12 am, and 6 pm ($N = 15\,992$). The main objective is to monitor the planets of our Solar System plus Pluto, the Sun, and the Moon, each with an importance value of 1. This way the maximum achievable point ($maxPoint$) is 10.

When noise is not considered the reward is simply given to the agent. Meanwhile, when seeking to speed up the training by using a hypothesis, the reward is further refined by (15). During the development, several hypotheses were tested, most of which were based on knowledge of the Solar System. It is known that the Earth's orbit around the Sun defines a plane $23.5°$ from the equator, called the ecliptic. Since the shape of the Solar System is more similar to that of a flattened disk, the planets move more or less in this plane. The orbits of these planets are at most $3°$ to the ecliptic, except for Mercury, which is $7°$, and Pluto, which is classified as a dwarf planet, being at an angle of $17°$. Using this knowledge, a hypothesis can be constructed that rewards placements that are not too far off the ecliptic.

We also know that for an observatory to see an object, it has to be within a cone defined by its AOV. Therefore, the closer the observatories are to each other, the higher the chance that their cones will intersect. We can specify a minimum distance or a minimum difference in either latitude or longitude and if the placement meets the condition, we can reward it.

The variable mini-batch approach described in section 2.1 does not change how rewards are calculated but uses a subset of generated positions instead as given in (13). Figure 2 depicts how the batch size $K$ changes during the training when using the parameters from Table 2. In the beginning, the

algorithm will only get rewards based on the chosen 64 samples, but it slowly increases to 15 992 according to (12) as discussed in section 2.

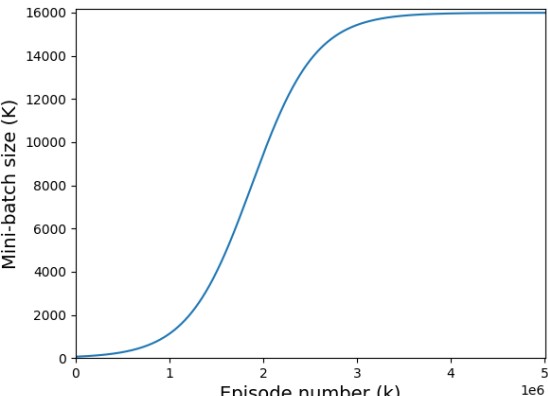

Figure 2: Mini-batch sizes $K$ used for the $k$-th episodes.

Table 1: Parameters used for adjusting the mini-batch size.

| PARAMETER | VALUE |
|---|---|
| $\epsilon$ | 0.0000009 |
| $\sigma_{max}$ | 40 |
| $T^{(0)}$ | 5 |
| $\alpha$ | 0.99997 |
| $N$ | 15 992 |

## 5 EXPERIMENTAL RESULTS

We have tried various heuristics, but in the end, $h(s_{t+1})$ was calculated in (15) as a combination of two different terms. The first gives a linearly decreasing reward for each observatory whose latitude value was below $20°$ with maximizing at $0°$, while the second rewards the algorithm if the longitude or latitude of a pair of observatories differs by at least $50°$. Intuitively, we have formulated that we expect the observatories close to the equator with a remarkably large distance from each other. For our experiments with $M = 3$ observatories we have fixed the maximum heuristics reward $\max_{s \in \mathcal{S}}(h(s)) = 3 \times (0.5 + 0.9) = 4.2$ in (15) empirically which is on a similar scale as $\widehat{r}_t$.

Using traditional PPO, the maximum reward achieved was $0.4568704$, which means that, on average, the observatories can monitor 4.5 celestial objects out of the ten chosen ones. Figure 3 plots how the rewards earned evolved during training. It can be observed that using heuristics the algorithm reaches the original PPO maximum slightly faster. The variable mini-batch size, however, shows a significant improvement in training speed, which can be further escalated by combining it with the hypothesis noise. As Table 2 more precisely indicates, we could speed up the learning process by almost $75\%$ in this way. The entries in the table were calculated by taking the elapsed times when the different approaches reached the maximum reward earned by the original PPO algorithm.

Table 2: Learning speed increments according to incorporating variable mini-batch and hypothesis.

| Method | Speed increase (%) |
|---|---|
| Hypothesis | 0.686 |
| Mini-batch | 70.409 |
| Hypothesis and mini-batch | 74.219 |

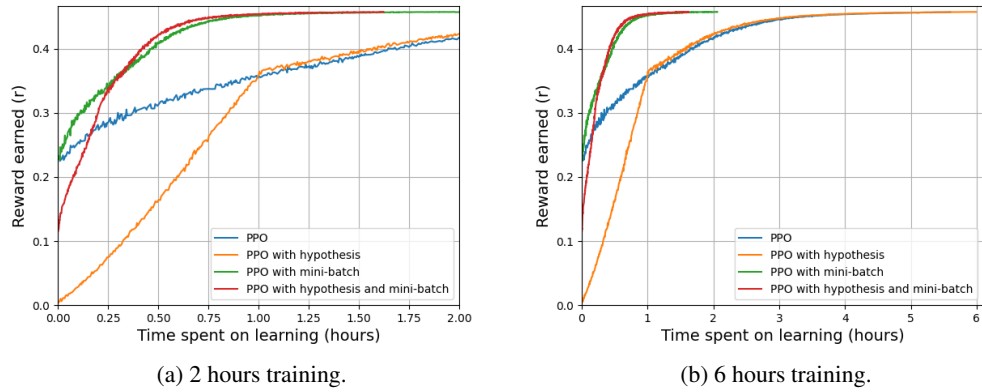

(a) 2 hours training.  (b) 6 hours training.

Figure 3: Comparison of the rewards earned (10-years period).

Using the hypothesis term solely did not increase the training speed remarkably in our above example. However, for a harder task including a 50-years time period with a capped maximum amount of steps and decreasing learning rate, it apparently outperformed the simple PPO. As depicted in Figure 4(a) the learning process reached the maximum reward with $47.099\%$ faster using the heuristics term. Figure 4(b) demonstrates also the optimal locations found for the 50-years example.

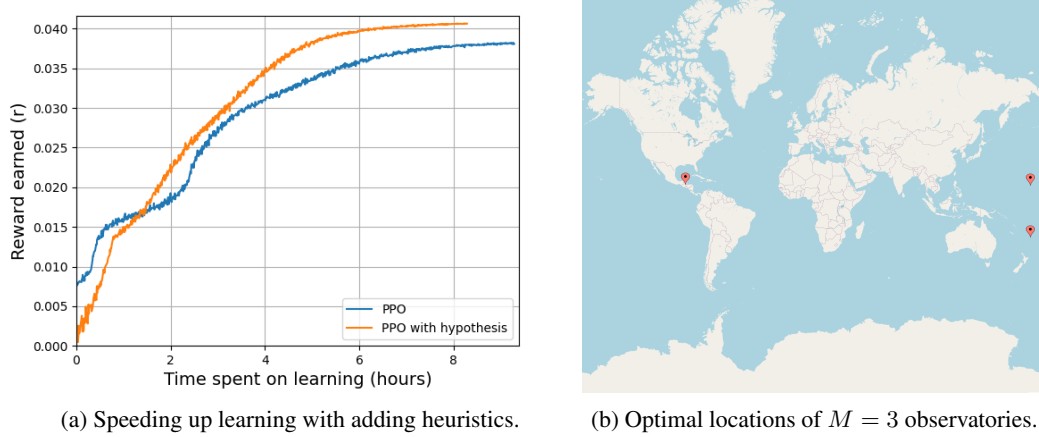

(a) Speeding up learning with adding heuristics.  (b) Optimal locations of $M = 3$ observatories.

Figure 4: An example for 50-years observation period.

## 6 CONCLUSIONS

In this work, we have introduced two new approaches, a heuristics term and variable mini-batch size, to speed up the PPO algorithm in RL. They have been incorporated as noises that vanish as the learning progresses to keep up the original convergence characteristics. We have demonstrated a remarkable gain in the training time in our experimental analysis focusing on finding optimal sensor placement to monitor space objects. Both of the approaches (esp. the variable mini-batch) were found to be efficient with an additional gain when they were combined. As a further step, we plan to let the sensors be placed in a more flexible way to other types of locations, which extension is expected to induce further modifications to our current solution.

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
