# OpenReview forum: "Speeding up Policy Optimization with Vanishing Hypothesis and Variable Mini-Batch Size"
_ICLR.cc/2023/Conference — Submitted to ICLR 2023_

### Official Review · Reviewer_XSaN · 2022-10-24

**Confidence:** 5
**Correctness:** 3
**Technical Novelty And Significance:** 1
**Empirical Novelty And Significance:** 1
**Recommendation:** 1

**Clarity, Quality, Novelty And Reproducibility:**

The paper is clear, but I question is novelty. It seem like another PPO paper that stacks small optimizations.

**Strength And Weaknesses:**

Strengths:
- Organized

Weaknesses:
- Writing is not formal (i.e. "takes a really long time", "remarkably reduce training time" in introduction)
- Evaluation is limited. To show the effectiveness of a method, must be applied to multiple environments (such as DMC control suite).
- Does your method also apply for different agents (outside of PPO)?
- Changing batch size/making it variable is not new. It has shown to be effective in supervised learning, so why not RL?
- Much of the evaluation, such as data acquisition, and learning process, should be moved to appendix
- Where is related work? Noise + variable batch size is not new.


**Summary Of The Paper:**

This paper introduces a novel method to drastically reduce training time for on-policy algorithms such as PPO. The formal method involves an exponential schedule for batch size and adding noise to reward. Their method is evaluated on an astronomical problem.

**Summary Of The Review:**

The paper has fundamental flaws and needs significant revision. If the authors can show the generality of their method on different RL agents and environments, I am willing to raise to weak reject.

---

### Official Review · Reviewer_RTtu · 2022-10-24

**Confidence:** 3
**Correctness:** 1
**Technical Novelty And Significance:** 1
**Empirical Novelty And Significance:** 1
**Recommendation:** 1

**Clarity, Quality, Novelty And Reproducibility:**

The clarity, novelty, and reproducibility all need substantial improvement.


**Strength And Weaknesses:**

**A minor weakness:** “incorporating some additional noise and performing noisy evaluation during the training process”.  In RL, this could mean many things, such as adding noise to the action selection or the policy parameters, noise to observations, etc.  From the abstract, we know that the authors mean to add noise to the reward function, but except for the phrasing in the abstract, this fact is not clear from sections 1-2 (there are plenty of phrases like the one above, which talk about adding noise, but exactly what that means is not made clear except for in the abstract).

**Major weaknesses:**
- T is first used in (3), but is undefined.  It is then used heavily throughout 2.1, but since it has not been defined, I cannot understand 2.1.
- “The second approach that we have realized is using a hypothesis as noise. To this end, we have decided on a fixed hypothesis h : S → R, where S is the state space that we have incorporated in the reward function r(s, a, s′) in the following way.”
    - small clarity problem: the hypothesis function and its codomain (the reals) are stated in the sentence above, but why is the codomain the reals?  What do the reals represent in this case?  It turns out the answer is that “h(s′) is the reward for arriving in the next state”, but this second quote is quite far below the quote above.  This makes it confusing and difficult to read.
    - larger clarity problem: h(s) maps states to rewards, but the details are unclear in Section 2.2.  Is it similar to a learned critic (predicting reward instead of return)?  Is it the true value of the next reward given the state and action?  Is it the expected value of the reward, given the state and policy?  Something else?  It is not clear from 2.2.
- There is no related work section.  The authors do not seem to have a firm understanding of the relevant related work in RL.  For example, the quote “converge faster than the vanilla policy gradient method (Mnih et al., 2016a)” may hint at a lack of breadth and depth knowledge about policy gradient methods.  There is a long history of policy gradient algorithms going back to Simple Statistical Gradient-Following Algorithms for Connectionist Reinforcement Learning (Williams, 1992) that I suspect the authors are unaware of.
- The experiments are not sufficient to support the authors’ claims.  There are no error bars, there appears to be only one trial per learning curve, and there is only one environment.  Unless if I missed it, there is no mention of hyperparameters or hyperparameter selection.

**Comments:**
- It seems like it might be trivial to formulate this problem as a supervised learning problem instead of an RL problem.  If that’s the case, treating this as an RL problem (without a *very* good justification) is not a great approach to the problem.
- One might argue that this is more of an applied paper.  Based on the title and abstract, I would argue that that is not the case/intent.  However, even for an application (astronomy) paper, most of the weaknesses above are still relevant.


**Summary Of The Paper:**

The authors propose adding noise to the reward function and a variable batch size to improve PPO.  They test their method on a simulated astronomy problem.


**Summary Of The Review:**

This paper needs some work before it is ready for a conference like ICLR.  I encourage the authors to bring more ML/RL knowledge to the project, and work on improving the writing, their knowledge of the related work, the experiments, and the overall contribution.

---

### Official Review · Reviewer_qi1M · 2022-10-25

**Confidence:** 3
**Correctness:** 2
**Technical Novelty And Significance:** 2
**Empirical Novelty And Significance:** 2
**Recommendation:** 1

**Clarity, Quality, Novelty And Reproducibility:**

The paper is poorly written, and appears to be of low quality and novelty. Reproducibility is affected by the lack of inclusion of code.

**Strength And Weaknesses:**

Strengths:
* The problem setting is interesting, and is a valid use case for RL.

Weaknesses:
* The paper has a significant number of shortcomings:
    * The idea of automatically tuning hyperparameters is not novel in RL. Indeed the entire sub-field of AutoRL [1] places emphasis on this, and even automatically discovering similar batch-size strategies to the ones here [2]. Furthermore, their notion of having a 'hypothesis term' designed to guide the reward is just a type of reward shaping and curriculum (e.g., in their case encouraging elliptical solutions). Again, no reference seems to be made to this fact.
    * There is no comparison of their method on standard benchmark suites, and no comparison to alternative scheduling methods.
    * It appears there's only a single seed used in experiments.

[1] Automated Reinforcement Learning (AutoRL): A Survey and Open Problems, Parker-Holder et al., JAIR2022

[2] Bayesian Generational Population-Based Training, Wan et al., AutoML2022

**Summary Of The Paper:**

This paper discusses the application of simulated annealing to the automatic tuning of hyperparameters in PPO, specifically applied to batch size and reward shaping (which they call a 'hypothesis term').

Evaluating on a sensor placing problem, which involves determining the optimal placement of observatory stations for the purposes of observing space objects, they see that they can converge to the optimal solution quicker than a standard PPO approach.

**Summary Of The Review:**

Overall the submission seems to be low quality, and doesn't merit acceptance. I think significant work needs to be done to improve this submission, and have given some guidance as to how to improve the paper along these lines.

---

### Official Review · Reviewer_Cmm4 · 2022-10-25

**Confidence:** 4
**Correctness:** 1
**Technical Novelty And Significance:** 2
**Empirical Novelty And Significance:** 2
**Recommendation:** 3

**Clarity, Quality, Novelty And Reproducibility:**

The writing could be improved, there are some grammar issues and typos. The motivation for the proposed heuristics, based on noise in simulated annealing, is also not clear. The mini-batch scheduling heuristic could on its own be interesting, if properly analyzed on more RL tasks (and there is no prior work on that in RL?). I don't understand the "hypothesis" reward noise heuristic, neither motivation, nor function.

Minor:
- "Reinforcement learning-based algorithms have been used extensively in recent years due to their flexible nature, good performance, and the increasing number of said algorithms." - Might be more likely that the popularity of RL is driving the number of algorithms than the other way around.

- "This is especially true in the case of policy-based algorithms which are notorious for being sample inefficient (Bastani, 2020). Although PPO, which is also a policy-based method, has better sample complexity (Schulman et al., 2017)" - At a glance, I'm not sure Bastani et al actually claims that. Note that sample "complexity" is usually asymptotic, I am also not sure if it has been proven that PPO has better such asymptotic complexity, although it tends to be faster in practice? Maybe rephrase this.

- On p.2 you cite (Minh et al, 2016) for what appears to be vanilla policy gradient

- "In our work, we focus on reducing this time by incorporating some additional noise and performing
noisy evaluation during the training process, " - Maybe you could rephrase this to better convey the gist of your idea - why would injecting more noise (PG is already very noisy) help convergence? I think the reason for noise in Simulated Annealing is to find a better global minima, but as you note yourself in many places, PG is a local approach that has no guarantees to converge to the global minima, and it is not at all clear that adding noise to the PG gradient will serve the same function as in SA.

- Did you run your experiments for multiple seeds? I see no confidence intervals on the curves

Some typos:
- " which approach can save a lot "
- " requirement to vanish the noise"
- "  till reaching  "

**Strength And Weaknesses:**

Strengths:
- Adjusting the mini-batch size is an interesting idea, and it has been proposed for (at least) supervised learning before.

Weaknesses:
- The overall mathematical rigor could be improved, and the writing is sometimes confusing.
- To test it only on your own application (and without error bars - did you make multiple runs?) makes it impossible to know if it will generalize to other problems. As a method paper, I would expect it to at least test on some gym environments (c.f. the PPO paper you cite).
- In more detail: I do not understand the motivation (or naming) of the second heuristic of injecting noise into the reward function that they call "hypothesis". The connection of both approaches to the temperature noise in simulated annealing is also not obvious. What I get is that the authors argue that since a decreasing noise sequence does not *impede* converge there, it will not here either. However, that does not mean it will *improve* convergence, and I don't understand how this is supposed  to work. In SA the noise is to find a better global minima, I do not think injecting it into the policy gradient will work as well. Is this what you are trying to do?



**Summary Of The Paper:**

The authors propose two heuristics for speeding up policy gradient approaches and demonstrate it with their application in observatory placement (a coverage problem). The first heuristic is a mini-batch size schedule, where they start with small batches. The second heuristic is injecting noise in the reward function.

**Summary Of The Review:**

While the mini-batch scheduling idea could be useful, the proposed theoretical motivation is unclear, and the experiments would have to be greatly expanded.

---

### Decision · Program_Chairs · 2023-01-20

**Decision:**

Reject

**Justification For Why Not Higher Score:**

The paper is definitely under the bar.
The contributions are weak and the presentation is poor.
A clear reject.

**Justification For Why Not Lower Score:**

N/A

**Metareview: Summary, Strengths And Weaknesses:**

The authors propose two heuristics to speed up the PPO algorithm.
The proposed solution is evaluated on a sensor placement problem.
Unfortunately, the reviewers found many weaknesses in this paper (minor novelty, poor presentation, single-run experiments, lack of baselines), which need to be addressed to make this paper ready for publication.
The authors did not provide answers to the reviewers' issues.
We encourage the authors to consider the reviewers' suggestions while preparing a new version of their paper.


**Summary Of Ac-Reviewer Meeting:**

N/A